# Museum as a Sensory Space: A Discussion of Communication Effect of Multi-Senses in Taizhou Museum

**Siyi Wang**

Department of History, Shanghai University, Shanghai 200444, China; wsy3192@shu.edu.cn; Tel.: +86-18905720103

**Abstract:** Museums are much more than repositories of cultural relics to be preserved for the future. They are centers of learning, community centers, social hubs—even places of healing and contemplation. The museum experience is a multilayered journey that is proprioceptive, sensory, aesthetic and social. In this context, this study takes the case of the 'People at the Seashore' multisensory area in the folk exhibition of Taizhou Museum, applies three data collection techniques (questionnaire, in-depth interview and observation) to assess various types of experiences (object, cognitive, social and introspective) and effects (visceral, cognitive and emotional) in the museum, and analyzes the practical effect and relative merits of the multisensory approaches used in this exhibition through the lens of communication effect. Accordingly, multi-senses acquire creative significances upon the attractive and holding power of museum exhibitions, specifically the emotional relevance and resonances. Thus, museums should be more concerned with the connection and complex interaction between senses and experience, meanwhile be active with visual, auditory, olfactory, taste and proprioceptive experiences and engage in the potential impact on visitors from cognitive and emotional aspects, which is an important trend for the museum's future development and also the vision of this study.

**Keywords:** museum experience; multisensory; evaluation; emotion

## 1. Introduction

Museums reach out to their communities by facilitating important and relevant conversations through their collections and by making the objects accessible and meaningful to a wide variety of visitors. Museum experience is a multilayered journey that is proprioceptive, sensory, intellectual, aesthetic and social. Besides, that makes a museum a place of learning, wondering, reflection and relaxation, sensory stimulation, new social ties, creation of lasting memories and recollection of past events. With the notion of 'intangible cultural heritage' was proposed, exhibitions have engaged more in presenting the information and stories behind the objects, that is, story-telling—especially the effort to represent phenomena. In the context of museum, Phenomena, which are more attend to the intangible, with the stories, people and time behind the objects per se, are different from objects. In museum, through the visualization and interpretation of information embedded in the objects, phenomena connect, arrange, express conforming to the logic of information, meanwhile, embed the storyline with abundant meanings into space, particularly, attaching the emotion into the context, consequently, witness the musealization from objects to memories, which is not only significant inside museum, but also towards the transformation of daily life with engaging connotation, ranging from a tiny portable object to a large city, people, things, objects and meanings that are related to the living world can be entirely musealized, creating prevalent memory museums. In this case, museum becomes not only a place that exhibits objects, but that allows visitors to experience, understand and even embody a story through those objects as well.

As we all know, various senses are consistently woven into people's daily lives. Therefore, a museum visit is more getting involved in interactions between senses and experience, such as visual, auditory, olfactory, taste and proprioceptive experiences and concentrates on the potential impact on museum visitors from cognitive, emotional and other aspects, all of which have evoked the multisensory museum to emerge.

In the context of multisensory museum, this study discusses how to apply senses in certain historical exhibition related to folk and its outcome. The research is based on the fieldwork conducted at Taizhou museum, Zhejiang (China) and shares the findings of analyzing the practical effect and relative merits of the multi-senses through the in-gallery questionnaire, in-depth interview, tracking observation and fix-point observation. This study also highlights the communication effect of senses through the lens of Kotler's [1] spectrum of museum experience and Doering's [2] museum experience types upon visitors' behaviors. Kotler's museum experience spectrum accords with the progressive effects of senses in museum which sheds light on the questionnaire of this research, primarily, it's the visceral experience from the stage of basic organs; Then it's the information and knowledge acquired through the visceral experience, videlicet, cognitive experience; Based on the cognition, senses, playing as triggers, activate the emotional memories and resonances. In the same time, if the spectrum of museum experience modal is step-by-step on depth, Doering's museum experience types are covered by range, of which the significances embrace from object, cognition to introspective and social experience, meanwhile enclose the purposes of visiting and by which the behaviors motivated, making a suggestive contribution to the observation of this research.

## 2. Literature Review

### 2.1. Multisensory Turns: A Study Based on the Senses

In the last four decades, the humanities and social sciences have twice witnessed the transformation of research orientation related to the mind and body. In the 1980s, 'the body turn' emerged [3,4], which stressed the fallacy of mind–body dualism and actively brought bodies back to the study of the humanities and social sciences. In the 1990s, 'the sensory turn' came into view, which highlighted that sensory experience is not merely about physiology, but also belongs to the historic and social studies [5,6].

'The body turn' encouraged the academic world to pay attention to mind–body topics and consider how to enable the 'body' back to research after 'interpretive turn' [7]. At the same time, the concept of embodiment, 'embodied mind' and 'embodied cognition' were introduced by George Lakoff and Mark Johnsen which emphasizes the formative role the environment plays in the development of cognitive processes. Embodiment as an effect where the body, its sensorimotor state, its morphology or its mental representation play an instrumental role in information processing. Literally, the abstract rationalities, in substance originate from the metaphorical projection of body experience (for more theory-driven accounts of how to define embodiment, see, e.g., Wilson, 2002; Goldman and de Vignemont, 2009), has been positively accepted by scholars, who have attempted to challenge the routine of regarding cultural activities as the 'mental' argument in the past. Instead, they are considered as 'embodiment culture' [8]. However, this approach also faces numerous challenges. Mind–body discussions involve complex philosophical consideration, and the initiation of 'embodiment' from phenomenology into the social sciences consistently requires redundant argumentations [9]. Thus, scholars may make huge efforts to present the body subjectively to emphasize the essence of 'the body turn'.

The sensory study has avoided redundant philosophical speculation about mind–body. As belonging to the category of 'bodies', the study of cultural and historical phenomena of sensory experience tends to present the significance of 'bodies' straightly. Interestingly, the most important contribution of 'the sensory turn' on epistemology does not lie in body emphasis but instead stresses a consensus that senses are not merely about physiology, they also intimate cultural, social and historical aspects [10]. This notwithstanding, five senses are commonly discussed separately [11] or

reviewing on 'ocularcentrism', which is being questioned in recent years. According to some scholars, perception refers to perceiving the surrounding world integrating different senses to receive information simultaneously, thus, we perceive our surroundings continuously and make the response [12–14]. As generally stated in the sensory-related books, single sense is specialized as the theme [15–18]. However, it's really difficult to work on the real-life of any culture by attending to a single sense, especially by stressing the crucial role one sense plays in a specific society [19] or by comparing the ratio of five senses adopted in different communities [20].

It follows that some sensory scholars have tried to propose modifications whose perspectives, however, still fail to sidestep the framework of considering individual senses as units of research [5,21]. We also find new words being used or invented by scholars, yet no details are provided. Words such as inter-sensoriality and multi-sensoriality are used by Howes and Pink, but they offer no explanation about how perception is formed by the multi-senses. The situation is similar for new words beginning with 'senses', such as sensory meaning [22], sensory interface [22], sensory harmony [23], sensory value [21], sensory symbolism [24] and sensorial landscape [25]. Although their meanings can be inferred from the word combination, precise theoretical definitions are not provided regarding perceptual nature, so the concepts behind these words are inadequate, though they often appear in articles as common sense. The use of these words without detailed definitions also suggests that there is a lack of a clear or clarified theoretical framework in sensory research.

### 2.2. Senses Interaction in Museums

In the humanities and social sciences fields, inspiring sensory research focuses on the diversity of sensory experience, typically on the five senses [20,26–29]. There are also studies examining how our sensory understanding changes throughout history and among different cultures [21,23,30], including much material of both general and particular interest related to museum history and museum experience [17,31–37]. An increasing number of studies of multi-sensory museums are being based on scientific evidence from neuroscience [38,39], and also begin a dialog between multi-sensory museum scholars and researches in neuroscience with producing innovative applications, devices, methods and sensory logistics [39,40].

### 2.2.1. Senses as Art

In the past few decades, artists have tried to integrate sound, smell, touch, action and even taste into their works [41–43]. Those innovative art pieces and art instruments have shed light on the exploration of the possibility of sensory art [44,45]. Multi-senses make itself as an exhibit and has gradually become the goal of some artists, which, in turn, challenges a museum's sensory application limitation to make it more diverse and experiential.

### 2.2.2. Senses as Information

With the 'sensory shift', modern museums have begun to reconsider their limitations to the sensory use of objects, and they are starting to explore the potential of multi-sensory solutions to improve knowledge transfer in museums and increase engagement with visitors by connecting them with the sensory properties of historic objects, their contexts and the stories behind, also providing emotionally enriched experiences. With increasing numbers of studies indicating that interacting with sensory objects have social, cognitive and even therapeutic value, especially for people with disabilities [46–52], museums are enhancing their valuation on 'touch' which is regarded as a therapy tool and a cultural communication platform [53–56], 'sound' which creates a sense of spatial experience [57,58] and 'smell' which triggers personal memories, imagination and emotion [59–61].

### 2.2.3. Senses as Phenomena

The phenomena show the contextualized living experiences with body and senses embedded in the museum object which was detached from original circumstances, de-contextualized and usually in dumb. As a result, the phenomenon world of museums is a portrayal of daily life which is full of

multi-sensory experiences, and people understand and experience the world through the senses and the body, under which thinking the use of multi-sensory in museums has gained more development opportunities. The phenomena embedded in the combinations can be either cognized or experienced so that they can "recall the power of expression" [62] on our senses more easily. Thus, multisensory immersion (immersion theory or flow theory is raised by Mihaly Csikszentmihalyi, which encompasses an experience of flow, through which a person performing an activity is fully immersed in a feeling of energized focus, full involvement and enjoyment in the process of the activity. Immersive experience is also examined by Privette and Bundrick, which is characterized as a process of inner enjoyment and shares similarities with the 'peak experience' and 'peak performance' named by Maslow) extends into the environment and the whole space of museum, typically by taking advantage of the latest virtual technologies [63–66]. Due to its prominent advantages and features in interaction, immersion and imagination, virtual technology is increasingly being used as educational tools in museums, on the bases of the multisensory learning which emphasizes the importance of direct experience and encourages learners to observe and experience the phenomena and rules of the real world using different kinds of senses [67–71].

The field of multi-sense museum research is very diverse and vibrant, but there are still many gaps in our understanding of how human senses can be completely engaged to gain maximum value from museum visits. As discussed above, most sensory studies and museum sensory research embark on the experience descriptions which turn out to be subjective and individual. Hence, this study attempts to investigate the preference and behavior of visitors from a more objective and general perspective through questionnaires and observations.

## 3. Material and Methods

Taizhou Museum, planned in 2004, officially opened on 12 July 2016, of which the third floor is themed 'Love for the earth: Taizhou Folk Customs from the Perspective of Cultural Geography.' This exhibition illuminates the evolution of Taizhou folk customs from ancient times to the present day concerning production, life, commerce and trade, convention and beliefs, through the display of masses of objects and dioramas divided into three parts: 'People in the Valley,' 'People by the Waterside' and 'People at the Seashore.' The 'People at the Seashore' exhibit displays the most typical fishing village in Shitang, Taizhou, which has been moved into the gallery and local village houses have been reconstructed. This exhibition extends the multi-senses-based experience, namely, the background of the fishing village, the sound of waves, the touch of sea breezes, the odor of fish mixed with the breezes and the flavor of small dried fish (Figure 1).

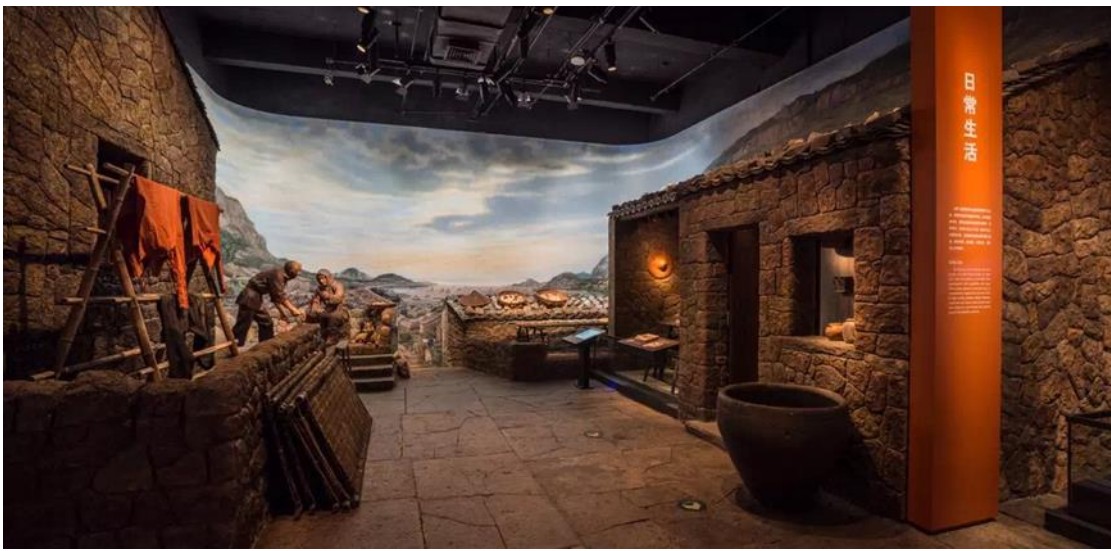

**Figure 1.** Area of Shitang village.

### 3.1. Curating and Interpreting the Multisensory Area

A field investigation was conducted in Taizhou Shitang fishing village, through which various photos of characteristic symbols of Taizhou seashore were collected. The broad horizon and offshore reefs, unique stone houses and fishermen's lifestyle are entirely visual affections of Shitang; while the sound of the sea waves crashing on the rocks, the whir of the sea wind and the calls of seabirds are the auditory symbols of the fishing village. During the curating and design process, the photos (Figure 2) collected from the ethnological fieldwork were used for the restoration of the scenes. For instance, the photos of the stone house, water vat, fishermen's food and bamboo basket for drying fish all came from the real daily life of Taizhou fisherman's family.

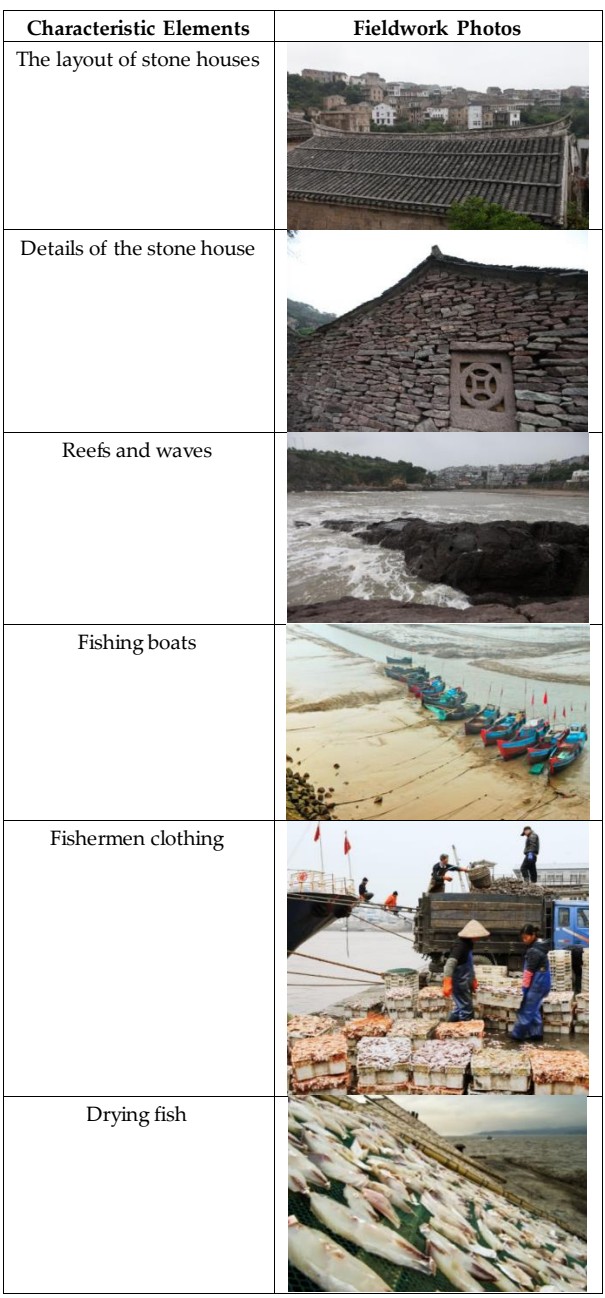

| Characteristic Elements | Fieldwork Photos |
|---|---|
| The layout of stone houses | |
| Details of the stone house | |
| Reefs and waves | |
| Fishing boats | |
| Fishermen clothing | |
| Drying fish | |

**Figure 2.** Materials of Taizhou fishing village from the ethnological fieldwork.

One of the considerable differences in the fishing village from other places is the sea wind, which is rather intense in Shitang. Thus, the whole village has the habit of drying their clothes near the

seashore. The fishermen who are exposed in the sea wind for long periods also show darker skin than people from other regions. Skin color was considered when designing the two fishermen statues, and a drying rack was placed near them to demonstrate the daily life of Shitang villagers. Furthermore, visitors can feel the sea wind by standing in the pathway of this area. Meanwhile, visitors are allowed to touch any exhibits of this area, both on display or in the dioramas. They can touch the walls of the stone house or put their hands into the water vat, which has engaged visitors to explore at will, and with pleasure.

Dried fish is a typical symbol of Shitang, of which a tasting setting exists for visitors to chew this special food. A strong, fishy smell left a deep impression on us when we took our first field trip to Shitang; this fishy smell never went away during our whole fieldwork. Therefore, this sense of smell, i.e., the fishy smell, forms an indispensable part and element of the impression on the fishing village. At the same time, the practice of adding smell to the gallery is particularly worth discussing. Increasingly, exhibitions are trying to involve a smell, which has indeed attracted more visitors, but meanwhile has faced a variety of challenges. Smell curatorship has certain obstacles in terms of selection of smell, placement, replacement, heavy or light, subjectivity and cost, etc.

"It is impossible to make a space completely odorless. No matter whether inside or outside a gallery, there are always smells from human activities or nature; so is the case with museums. The odor molecules of food or drinks such as coffee spread to every part of a space; the scent of perfume from visitors is also left in the gallery."(Interviews with Kaichen Zheng and Junzhu Liu from the design team in Hangzhou Obsidian Exhibition Design Co., LTD, 27 July 2016) Indeed, temperature, humidity, airflow and size of the gallery are basic factors that impact the state of smell of a space. At the same time, smell is a highly subjective sense, which is one of its unique features different from other senses. Different people with different experiences may impact different personal relevance and judgments on a specific smell, which seems to make the "smell-sharing" inaccessible. In the 'People at the Seashore' area, some visitors are very resistant to the fishy smell and would not step into it. However, visitors who used to live by the sea enjoyed the smell naturally. "An odor generator using fine chemicals was adopted to ensure the safety of the fishy smell and control its injection volume. However, the cost of developing smell is rather high in the beginning, but it will drop as the smell is made into a product and is widely used in the market."(Interviews with Haodong Zhang and Dongshen Zhan from the facilities team in Hangzhou Obsidian Exhibition Design Co., LTD, 1 August 2016) In addition, it is also critical to avoid visitor olfactory fatigue in multisensory exhibitions, which turns out to be even more dramatic than other museum fatigues, as people's whole olfactory systems will shut down if olfactory fatigue happens. Hence, adding smell in a exhibition is a bright new attempt, yet simultaneously, a formidable challenge.

### 3.2. Visitor Studies in the Multisensory Area

Visitors can enjoy a multi-dimensional impression of the seashore during their first visit to this area in the exhibition, and they can also arouse the empathy with the Taizhou natives. The more senses that are involved, the more the restored scenes will appear closer to real daily life. With continuous innovations in exhibition technology, multisensory approaches have been increasingly used in exhibitions to deliver a more striking and realistic immersive experience to visitors through time and space. However, some questions may arise: How about the effect of multisensory approaches on visitor cognition and learning? How do visitors explore their multisensory experience? How to address the multi-senses in the exhibition?

Based on the above inquiring, this study raises the following questions to be investigated and clarified:

- How multi-senses influence the behaviors of museum visitors?
- What's the effect respectively of multi-senses upon the visceral experience, cognitive experience and emotional experience?
- What's the role of multi-senses plays in the overall experience rate?

This research takes Kotler's Museum experience spectrum, Doering's Museum experience types on the visitors' behaviors and Pekarik et al.'s IPOP theory [72] with experience preferences as data instruments. The museum experience is divided into visceral experience, cognitive experience and emotional experience as stated by Kotler, which have their sub-questions in detail (Table 1);

**Table 1.** Museum experience spectrum [1].

| Visceral | Emotional | | Cognitive |
|----------|-----------|-----------|-----------|
| Excitement | Playfulness | Contemplation | Learning |
| Adventure | Transfer | Dream | Exploring |
| Fantasy | Game | Reflection | Testing |
| Immersion | Sports | Aesthetic | Analyzing |
| Novelty | Social | | Recognizing |
| | Shopping | | Skill |

While Doering presented four museum experience types conforming to visitors' behaviors, namely, object experience, cognitive experience, introspective experience and social experience.

Social experience centers on one or more other people, besides the visitor.

Object experience gives prominence to the artifact or the "real thing."

Cognitive experience emphasizes the interpretive or intellectual aspects of the experience.

Introspective experience focuses on the visitor's personal reflections, usually triggered by an object or a setting in the museum. (p. ii, [2]).

Furthermore, Pekarik and his colleagues categorized experience preferences into idea preference, people preference, object preference and physical preference, as a predictive modal for visitors' behaviors.

Ideas—an attraction to concepts, abstractions, linear thought, facts and reasons; People—an attraction to human connection, affective experience, stories and social interactions; Objects—an attraction to things, aesthetics, craftsmanship, ownership and visual language; and Physical—an attraction to somatic sensations, including movement, touch, sound, taste, light and smell. (p. 6, [72]).

In-depth interviews were given with the curators (see Appendix A), the design team (see Appendix B) and visitors (see Appendix C) with respect to the 'People at the Seashore' area in the folk exhibition. A survey among the museum visitors through fixed-point observation, tracking observation and questionnaires. The pre-survey was conducted on 14 July 2016; while the formal one was conducted between 10 August 2016 and 17 August 2016 (but not on 15th, Monday, when the museum was closed); and the fixed-point observation and questionnaire were carried out between 9:00 am and 11:30 am and from 2:00 pm to 4:00 pm. The survey (Figure 3) is covered from Point A to Point B in the 'People at the Seashore' area, the observation began at Point A and ended at Point B the questionnaires were given at Point B.

Since the Taizhou Museum has much daily traffic flow, the fixed-point observation and CCTV tracking were used in this research, with selecting targeted visitors and filling in the observable basic data, e.g., gender and estimated age group, if accompanied. To record visitors acting naturally, CCTV was used to clearly record their behaviors. Questionnaires were completed at Point B with the consent of visitors.

Three observation points (Figure 4) were selected: Spot 1 (Figure 5) is the food diorama of the fishing village, Spot 2 (Figure 6) is the sea background landscape and Spot 3 (Figure 7) is the fishermen statues and clothes-drying diorama.

In the visitor fixed-point observation grids, each point was designed separately. Each group was provided with a card with the grid and the total number at each point was counted according to the numbers of cards after the survey. The observed visitors were randomly selected. The sample size was 893 groups of visitors totaling 2152 individuals. The 'Remarks' column of the observation grid was used to record the observed visitors' bending and touching at exhibits, the intentional changes in

viewing angles, obvious acceleration/deceleration and talks, among other data. 'Route' was filled with descriptive records that appropriately reflected the general route in the observation area. 'Speed' was measured in 'm$^2$/min,' calculated against the time the observed visitor remained in the predefined area, which was used to reflect the influence of a specific sense on a visitor's behavior.

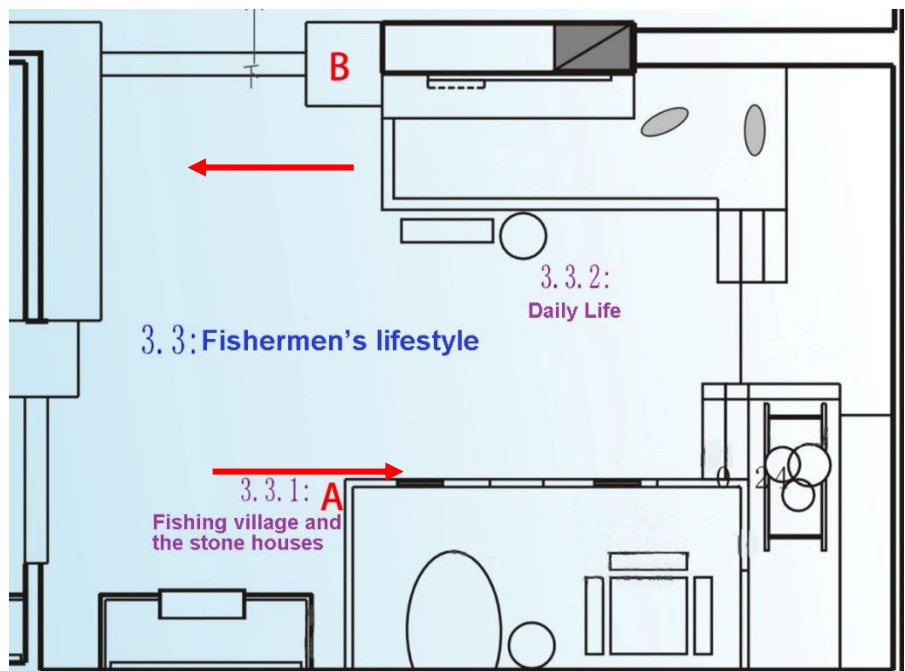

**Figure 3.** The observation area.

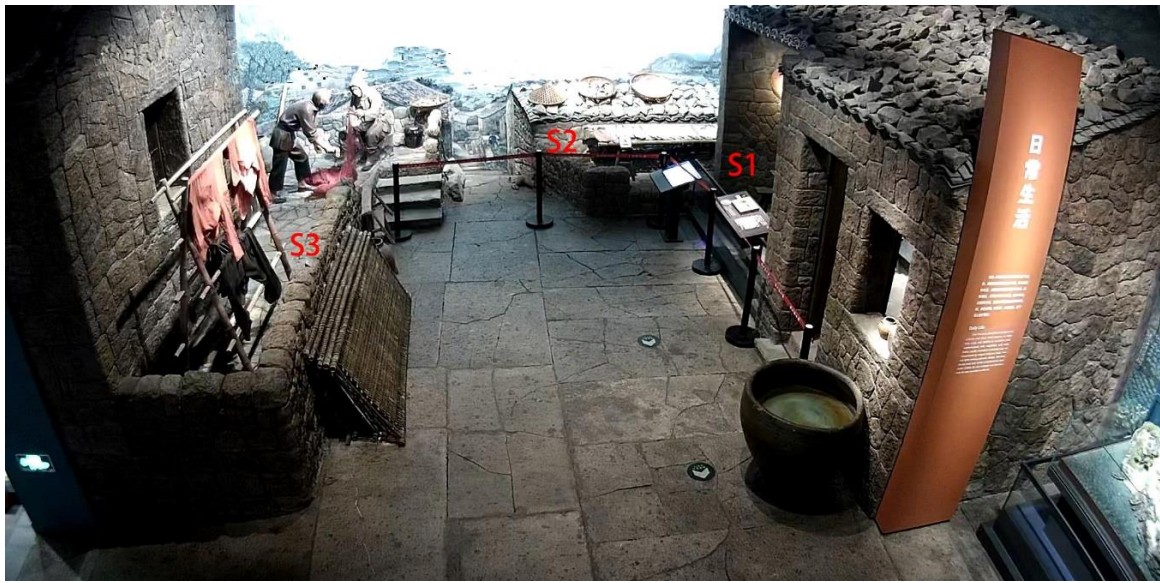

**Figure 4.** S1, S2 and S3 observation points.

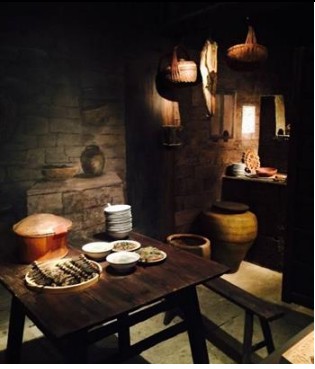

**Observation Indexes and Behavior Records**
**（Food Diorama/Label Screen）**
**(Y is the abbreviation for Yes, and N is for No)**

| Stop | Stay Time | Take Photos | Bend Body | Lean | Remarks |
|---|---|---|---|---|---|
| Y/N | | Y/N | Y/N | Y/N | |
| Touch exhibits | | | Routes | | |
| Y/N | | Straight/walk to **label screen** /walk to **food diorama** | | | |
| Type | （Single/Group/Others） | Gender | （Male/Female） | Age | （Youth/Mid-age/Aging Speed |

**Figure 5.** S1 (Fishermen's food diorama) observation grid.

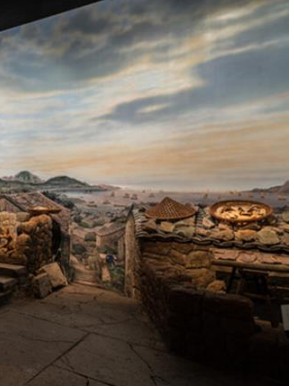

**Observation Indexes and Behavior Records**
**（Background Panorama/Restored Diorama /Wave Sounds）**

| Stop | Stay time | Take Photos | Look Around | Stoop | Look Above | Remarks |
|---|---|---|---|---|---|---|
| Y/N | | Y/N | Y/N | Y/N | Y/N | |
| Touch exhibits | | | Routes | | | |
| Y/N | | Straight and look/Ignore and leave | | | | |
| Type | （Single/Group/Others） | Gender | （Male/Female） | Age | （Youth/Mid-age/Aging） Speed | |

**Figure 6.** S2 (Background of the fishing village) observation grid.

The questionnaire (see Appendix D) includes two questions: the basic information of the visitors and the visiting effects. Question 1 (Q1–Q6) covers six items: gender, age group, education background, if local or not and if the visitor has ever visited 'People at the Seashore' before. Question 2 (Q7–Q18) is a scoring scale that includes the overall experience of the exhibition, the effect of sensory elements, information contained in the exhibition and acceptance of the multisensory exhibition. Scores in each question range from one to five. The questionnaire was inspired by Kotler's spectrum of museum experience to divide visitor experiences into visceral, emotional and cognitive ones [1]. A total of 150 questionnaires were distributed at the exit of 'People at the Seashore' area. All questionnaires were returned (a rate of 100%); 148 were valid (a rate of 98.7%).

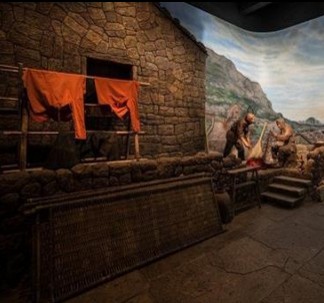

**Observation Indexes and Behavior Records**
**（Restored Diorama /Sea Breeze）**

| Stop | Stay Time | Take Photos | Bend Body | Look Above | Remarks |
|------|-----------|-------------|-----------|------------|---------|
| Y/N | | Y/N | Y/N | Y/N | |
| Touch exhibits | | | Routes | | |
| Y/N | | | Straight and look/Ignore and leave | | |
| Type （Single/Group/Others） | Gender （Male/Female） | | Age （Youth/Mid-age/Aging） | Speed | |

**Figure 7.** S3 (Fishermen's clothes diorama) observation grid.

With the exception of Q7, which is about the overall experience rate, Q8–14 concern visceral experience, whether the visitor feels any sensory elements or not; Q15–16 concern cognitive experience, whether there are any information and understanding; and Q17–18 concern emotional experience, whether there is any memory or emotion. The analysis was based on the above three aspects, calculating with the mean scores for each aspect which indicate the effect, then the correlations between the mean scores of Q8–14, Q15–16, Q17–18 and gender, age, education, Taizhou local or not, ever been to the fishing village were analyzed applying the independent sample T-test with the level of statistical significance known as *p*-value. A *p*-value less than 0.05 (typically ≤ 0.05) is statistically significant, which indicates strong evidence against the null hypothesis, as there is less than a 5% probability the null is correct. The overall experience rate and the mean scores of Q8–14, Q15–16, Q17–18 were calculated applying one-way ANOVA with the hypothesis that the strong association between these overall ratings and visceral (Q8–14), cognitive (Q15–16) and emotional (Q17–18) experience. In this research, the data were collected and analyzed using SPSS 21.0, unreasonable values were deleted, and the missing values were replaced by the mean values.

## 4. Results and Discussion

### 4.1. Length of Visiting Time

Among the 893 groups, the average length of visiting time from entering this area (Point A) to leaving (Point B) was 81.4 s, or approximately 1 min and 20 s; the longest visiting time was up to 273 s (or approximately 4 min and 30 s) by a middle-aged couple. The author counted those visitors visiting longer than 200 s, and most were middle-aged or aging visitors, of which the vast majority were couples or families; the shortest visiting time was 13 s by a little girl who complained of the offending odor while taking photos, and left covering her nose. During the observation, families spent the longest visiting time; most of their time was used to educate their children in how to behave or to interact with them; group visitors, mainly visitors from several companies (>3), spent the shortest visiting time; for instance, several groups of students (20–30) were guided by their docent, and their visiting time was similar, approximately 56–77 s, which was shorter than average.

With regards to the behaviors of visitors who spent more than the average visiting time (81.4 s), according to Doering's four museum experience types: object experience, cognitive experience, introspective experience and social experience [2], we found that the behaviors of visitors who spent more than the average visiting time included almost all four kinds of experiences (Table 2).

**Table 2.** Behavior table of above-average visitors.

| Behaviors | Experience Types | Percentages (Percentage of Above-Average Visitors) |
|---|---|---|
| Touch exhibits (water vat, fishing net, dried fish) | Object experience | 60.5% |
| Talk and recall memories | Social experience & Introspective experience | 55.3% |
| Take photos | Social experience | 50.0% |
| Bend body to observe objects | Object experience & Cognitive experience | 42.1% |
| Read the labels | Cognitive experience | 31.6% |
| Question and discuss | Cognitive experience | 22.4% |
| Look above | Object experience | 14.5% |
| Look around | Object experience | 13.2% |
| Touch the wall of stone houses | Object experience & Introspective experience | 10.5% |
| Read the touch screen | Cognitive experience | 4.0% |
| Squat to observe objects | Object experience & Cognitive experience | 4.0% |
| Feel sea breezes with hands | Introspective experience | 2.6% |

*4.2. Attractive Power and Holding Power*

Points S1, S2 and S3 were selected as the observation points, namely, the attention points attracting the visitors to remain for a while. S1 mainly introduced the food of the fishing village and reproduced the food on the table in the houses, with text labels and touch screen. S2 was set in a large seashore painting, with seabirds chirping, reproducing the daily life of fishermen, including drying net, fish and the unique fish odor. S3 showed the fishermen's clothing, including statues of two fishermen and the effect of sea breezes.

Visitors stopped at the attractive three points for different lengths of time, as different objects and dioramas varied in holding power. Holding power is a quantitative indicator of visitor interest that refers to the average time spent of visitors who were attracted by a specific object.

It can be expressed by an equation:

$$\text{Holding Power} = \frac{Total\ effective\ time\ spent\ at\ a\ particular\ object}{Sample\ size\ of\ effective\ visitors} \tag{1}$$

wherein the sample size of effective visitors refers to the number of visitors who spent at a particular object more than 3 s, and total effective time refers to the total time that the visitors spent at that object.

According to the statistics, we found that the holding power at S1 was greater than that at S2 the holding power at S2 was greater than that at S3, briefly, S1 > S2 > S3. This finding can be attributed to the IPOP model proposed by Pekarik, the visitors' preferences to the museum exhibition can be divided into four types conforming to their original interests, i.e., I = Idea, P = People, O = Objects, P = Physical [72]. These four types of visitors will be attracted by the factors in the exhibition depending on their preferences. Therefore, if a display area has elements of all four types, it will definitely arouse the interests of most visitors.

S1 was provided with illustrated panels to meet the needs of idea visitors, including a touch screen that told about the food story of fishermen in Shitang to meet the needs of people visitors; a restored diorama with objects was provided to meet the needs of objects visitors; the encouragement for touching was to meet the needs of physical visitors. At S2, chirping seabirds, fish smell and touchable dried fish met the needs of physical visitors, but visitors varied in their acceptance of fish smell; some left the gallery because of the offensive smell. At this point, visitors were not provided with text or story narrative, and the objects were much less diverse than at S1; S2 attracted fewer visitors and had a smaller holding power than S1. During the observation, more visitors simply ignored S3 and directly left this area, as it was also not provided with text or story narrative, and almost no obvious object was displayed. In terms of physical experience, the effect of the breeze was not satisfactory and rarely visitors felt it, while the touchable bamboo rack placed to the side was often regarded as a part of the diorama which fails to draw attention. Furthermore, it was darker here than at the other two points.

The results of these three multisensory attractive points suggest that even the physical experience, with its higher multisensory engagement, may not meet the needs of visitors other than physical

visitors. Thus, the multisensory area should also consider adding illustrations, personal stories and a variety of objects. However, compared with any other monosensory exhibition, multisensory one gives full play to physical visitors' initiatives and involve with more visitors.

*4.3. Questionnaire*

### 4.3.1. Visceral Experience

Q8–14 focused on the visceral experience through the tactile, auditory, olfactory and visual senses; the evaluation criterion was whether these senses could be acquired. By our findings, the average score of Q8–14 was 3.91. Aside from three items—the olfactory sense, the feeling of the fishing village and whether it was similar to a Taizhou fishing village—the remaining items were lower than the average score (Figure 8). This reflects the visitors' relatively low degree of identification with other senses, except for the smell and sight. It was also clear that the influence and effect of other senses on one's visiting experience were relatively weak. Through the correlation analysis, age and whether the visitors had been to a fishing village showed a significant correlation with the scores of Q8–14.

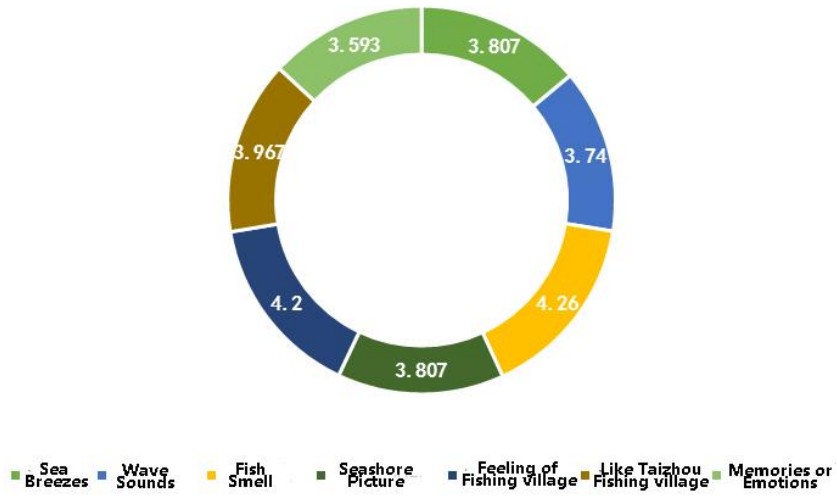

**Figure 8.** The average rate of visceral experience.

### 4.3.2. Cognitive Experience

Q15–16 mainly focused on visitors' understanding and the information about their impression of a fishing village. The average score was relatively high: 4.28, which suggests that the interpretation and presentation through multi-senses enhanced visitors' understanding, while simultaneously building a stronger impression. Meanwhile, we noticed that whether visitors were Taizhou locals or not showed no significant ($p = 0.69$) on cognition experience. That is to say, prior experience was not an influential factor of senses to some extent, the effect of which upon cognitive experience was rarely limited by regions and prior knowledge. This turned out to be the advantage of learning and understanding, expressly: every individual embraced the specific talent, which was to experience object with senses and body, cultivated by daily life.

### 4.3.3. Emotional Experience

Q17–18 reflect visitors' attitude and verify whether multi-senses can stimulate emotions and memories. The average score was 4.11. In terms of attitude, visitors held supportive opinions. Meanwhile, stimulation of emotions and memories showed a significant correlation with age ($p = 0.001$) and whether the visitors had ever been to a fishing village ($p = 0.022$) (Figure 9) and demonstrated a significant positive correlation with age. In other words, older visitors were inclined to have memories, emotions and a retrospection of the past triggered through the multi-senses of the fishing village.

In addition, for those with the experience of visiting a fishing village who gave me some specific feelings during the survey and interview—for instance, 'I remembered and cherished the childhood in Yuhuan, which is another fishing village in Taizhou.'—the effect of recalling memories and triggering emotions was significant. Thus, dedicating to the specific and realistic details of senses would further motivate visitors' understanding and emotional relevance.

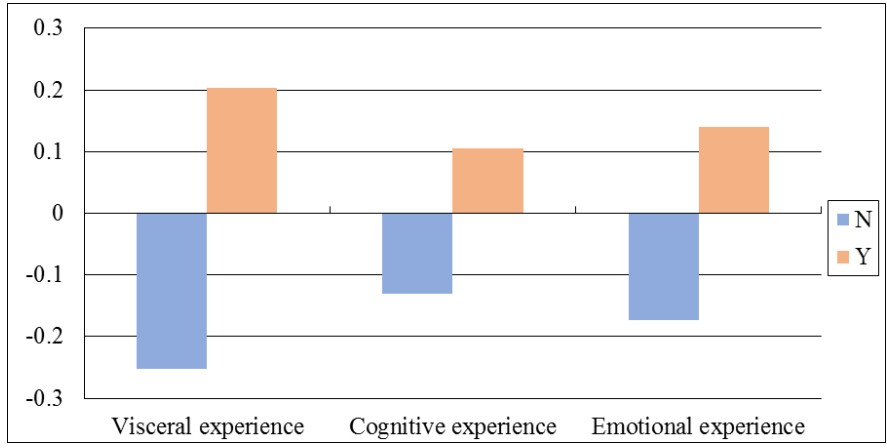

**Figure 9.** *t*-test for whether or not visitors have ever been to a fishing village.

### 4.3.4. Overall Experience

The scores of the overall experience in Q7 were all from 3 to 5. In other words, no visitor rated 1 or 2. Among the visitors who rated 3, 4 and 5, a statistical significance and a certain tendency was seen in Q8–14 (visceral experience, $p = 0.000$), Q15–16 (cognitive experience, $p = 0.000$) and Q17–18 (emotional experience, $p = 0.000$). Visitors who scored low in these three aspects of questions also did so in overall experience rate, that is, they rated 3 or 4 which was below the average. Visceral experience was the most effective in reflecting the differences between visitors who rated different scores. This finding indicates that the polarization of visceral experience exerted a direct influence on one's overall experience (Figure 10).

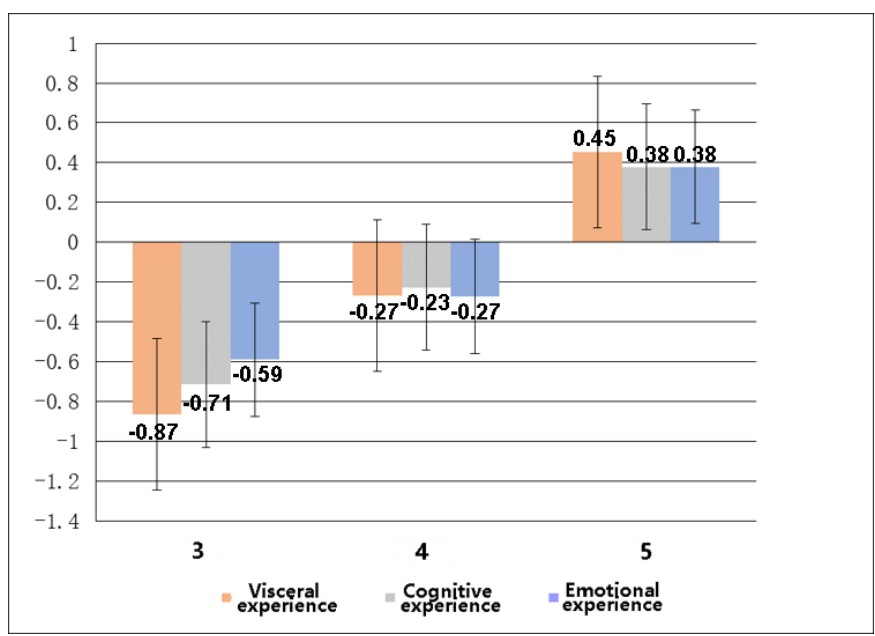

**Figure 10.** The three-type test for overall experience rating.

The above investigation indicated that the engagement of multi-senses can increase the attractiveness of the exhibition to visitors and prolong their visiting to a certain extent. The intuitive sensory experience not only facilitated visitors' acquisition of cognitive knowledge and information but also enhanced the emotional connection and resonance among specific visitors. The visceral experience examined above, to a large extent, determined the satisfaction of visitors with their experience in the museum, and the emotional connection also contributed to their overall experience. The merits of multi-senses can be directly presented through questionnaires. However, during the observations of visitors' behaviors, we found that using multi-senses solely without the engagement of other approaches such as text, pictures and multimedia could influence the function of knowledge transfer. Thus, the best effect may be achieved only when multi-senses are combined with other presentation approaches. Furthermore, the instability of sense properties, such as smell, also had a certain influence on the accuracy of measurement and observation. Nevertheless, the creative role of multi-senses in the exhibition was remarkable.

## 5. Conclusions

Museums in the 21st century have witnessed the various senses playing increasingly instrumental roles in visitors experiences. For the multisensory area in the Taizhou museum, analysis of the communication effect of senses revealed encouraging prospects for engaging multisensorial exhibits. Despite the effect of multisensory exhibits in attracting and engaging visitors, senses vary strongly with individuals, hence visitors who have different sensory "positions" show different responses, which makes this research rather difficult and inadequate. In consequence, through data analysis, we cannot identify the accurate impacts of multisensorial exhibits in the museum on individuals who have different acquisition abilities of the senses, different sensitivities and different personal experiences. Despite this, the common consensus is that senses have a direct impact on visitors' satisfaction, which is more obvious when engaged with emotional connections for multisensorial exhibits that can trigger the emotional resonance of visitors. The multisensory design of museum exhibitions is not merely confined to superficial designs for enjoyment, but more as a catalyst and an intermediary which plays a basic, but critical role in emotional engagement, reminiscence and personal relevance and reflection.

From being a gallery of 'single-sense epiphanies' [73], the museum is being transformed into a sensory gymnasium [34] which is attempting to break the limitations of sight to help museum visitors understand and appreciate more about art, history and culture. What it emphasizes is that visitors— through their multi-senses—may retrace museum objects, phenomena and culture diversely, to sense the connection between people and the world. The multisensory practices in the museum start from a meaningful theme and adopt specific sensory "props" to create a contextual space—while simultaneously creating a personalized narrative intimately attach to multi-senses, thus shape a vivid and impressive sensory experience.

In this study, we argue that the multisensory museum is not merely a multi-senses feast. Furthermore is the sensory connection, which turns out to be enlightened, significance-exploring and intriguing, through the engagement of multiple senses, including visual, auditory, olfactory, taste and proprioceptive experiences, that plays an instrumental role in creating an immersive experience that stimulates emotion, reminiscence and education. Through the media of multi-senses, a museum visit is improved by delivering cultural-ethical enrichment, which is the mission and vision for masses of museums.

**Funding:** This work was supported by The National Social Science Fund of China under Grant Number 19ZDA174.

**Acknowledgments:** This study includes significant contributions from the Taizhou Museum Director, Yuhong Lao and Hangzhou Obsidian Exhibition Design Co., LTD staff: designers Kaichen Zheng and Junzhu Liu, fabrication team Haodong Zhang and Dongshen Zhan. Thanks to my supervisor Jianqiang Yan for supporting me in carrying out the museum survey. Finally, special thanks go to Zahava Doering, Andrew Pekarik and Kym Rice for sharing their ideas about the multisensory museum.

**Conflicts of Interest:** The author declares no conflict of interest.

**Appendix A. In-Depth Interview Questions with Curators**

1. The folk exhibition in Taizhou Museum presents various lifestyles in the different landforms in Taizhou, i.e., the mountains, the plains and the seashore areas. Of these exhibits, the multi-sensory seashore area is an indispensable part of the exhibition. What made you add this area to the exhibition? If this area can be regarded as an approach, what is its role in the exhibition? Is it helpful in achieving the mission of the exhibition?

2. The multi-sensory seashore area is widely recognized by visitors in making the exhibition more enjoyable. was enjoyability one of your intentions when you added this area? What is the major purpose of adding the multi-sensory area?

3. As we know, our senses are crucial and closely related to our memory. When adding this area to the exhibition, are you attaching more efforts to evoke the memories of visitors and leave them with a deep impression of the multi-sensory context with their own experiences and feelings?

4. What do you think are the pros and cons of adding multi-sensory areas to the museum?

5. Do you think multi-sensory-related projects can be popularized to other museums? What factors do you think would influence museums to usher in multi-sensory projects?

6. In your previous interviews, you mention that, "four of the five senses have been presented, and the last sense of taste can be also presented if the museum provides a unique seaside drink". So, in the future, will you update the formation of this multi-sensory area?

7. Have visitors' experiences of this multi-sensory area been in line with your expectations?

8. What is the possibility and prospect of multi-sensory-related projects in the future?

**Appendix B. In-Depth Interview Questions with the Design Team**

1. Have you engaged in multi-sensory-related projects in museums before?

2. How do you present the five senses in this multi-sensory area? What technical support do they need?

3. What were the major steps to complete the design of this multi-sensory area? How was this area made?

4. Since the multi-sensory area involves different senses, what should be considered about the space and scale during the design?

5. The multi-sensory program may attract more visitors to gather together. Are there any considerations about exhibition space planning? How do you arrange the space?

6. What is the cost to produce a multi-sensory area like this? Do you think such multi-sensory-related projects can be popularized to all museum exhibitions?

7. What are the limitations of designing a multi-sensory-related project? How can you overcome them to some extent?

8. What is the possibility and prospect of multi-sensory-related projects in the future?

**Appendix C. In-Depth Interview Questions with Visitors**

1. How do you feel after visiting this area?

2. How can one make this area more relatable, more realistic and more interesting than it is now, in your opinion?

**Appendix D. Visitor Questionnaires**

1.Gender: ▫Male ▫Female

2.Age: ▫15 and below ▫16—25 ▫26—35 ▫36—45
    ▫46—55 ▫56—65 ▫65 and above

3.Education: ▫Middle school (and below) ▫High school ▫College ▫Undergraduate ▫Graduate (and above)

4. Are you Taizhou local? ▫Yes, ________(Area) ▫No, I am from________(Province/Nation)

5. Have you ever been to a fishing village? Have you had a certain experience? ☐Yes ☐No

6. Did you visit the entire Fishermen's Daily Life area? ☐Yes ☐No

| NO. | Questions | Rank by extent from 1–5 | | | | | Notes |
|---|---|---|---|---|---|---|---|
| 7 | What is your overall experience in this area? | Boring ⟶ Interesting | | | | | (Don't know or not pay attention etc.) |
| | | 1 | 2 | 3 | 4 | 5 | |
| | | None ⟶ A lot | | | | | |
| 8 | Did you feel the sea wind in this area? | 1 | 2 | 3 | 4 | 5 | |
| 9 | Did you hear the sound of waves in this area? | 1 | 2 | 3 | 4 | 5 | |
| 10 | Did you smell the fish odor in this area? | 1 | 2 | 3 | 4 | 5 | |
| 11 | Did you see pictures of the sea in your mind? | 1 | 2 | 3 | 4 | 5 | |
| 12 | Is this area giving you the sense of a fishing village? | 1 | 2 | 3 | 4 | 5 | |
| 13 | Is this area like a real Taizhou fishing village? | 1 | 2 | 3 | 4 | 5 | |
| 14 | Does this area evoke any certain memory or feeling? | 1 | 2 | 3 | 4 | 5 | |
| | | Strongly disagree ⟶ Strongly agree | | | | | |
| 15 | I think this area makes it easier to understand Taizhou seashore life. | 1 | 2 | 3 | 4 | 5 | |
| 16 | I think this area deepens the impression of Taizhou seashore. | 1 | 2 | 3 | 4 | 5 | |
| | | Strongly disagree ⟶ Strongly agree | | | | | |
| 17 | I think it is easy to recall memory and get inspired by this area. | 1 | 2 | 3 | 4 | 5 | |
| 18 | I hope the museum can use the same approaches as this area into more exhibitions frequently. | 1 | 2 | 3 | 4 | 5 | |

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
