# Peer review of "Museum as a Sensory Space: A Discussion of Communication Effect of Multi-Senses in Taizhou Museum"

_sustainability, doi:10.3390/su12073061_

Round 1
Reviewer 1 Report
The word ‘interpretation’ in the fields of museum & heritage has a special meaning, i.e. explaining an object & its significance.
Objects are made meaningful according to how they are placed within relations of significance, and that these relationships depends on who is determining what counts as significant. Objects are used to construct the past, but never possible to reconstruct the past entirely. Objects are used to construct identities, on both personal/ national level.
The application of multi-senses and story-telling approach in Taizhou Museum is a kind of interpretation of its fisherman-related objects/ collection by the exhibition curator.
If it is a paper in the fields of museum & heritage, it is worth to add in related discussion.

Reviewer 2 Report
The paper draws upon the sensory and embodiment turns in social and humanistic sciences. Using two marketing frameworks for classifying the museum experience (Doering, 1999; Kotler, 1999), it applies three data collection techniques (oral survey, in-depth interview, and observation) to assess various types of experiences (object, cognitive, social, and introspective) and effects (visceral, cognitive, and emotional).
Normally I would reject the paper based on its multiple faults (see below). However, it might deserve a chance as it has the potential to be of interest to the journal`s readers due to the actuality of the topic, the variety of data collection techniques, the complexity of applying the observation in museums` settings, and the quality of collected data. Another strong point is that the Literature review raises very interesting ideas, but unfortunately they are not further developed throughout the paper and their connection with the Results is missing.
In brief, there is no coherence between various sections of the paper and no clear argument developed throughout the paper (also since there is no clear information about the research question and hypothesis, besides the vague aim ”to discuss how to apply senses in certain historical exhibition related with folk and the outcome”). The Literature review does not justify the choices made to design the study and is not properly linked with the results and their implications. The author does not convince to be familiar with the norms of reporting research results through peer-review articles. For instance, only one data collection instrument is described in the Material and Methods section, another one is introduced in the Results while the third one is not explicitly presented (which make the study hard to reproduce). Similarly, although the author assumes in the Introduction to use only one theoretical framework (line 48), in the Results, a second one is introduced (line 215) without any given reason. The reporting is rather unintelligible for the readers which have to move between various sections of the paper to understand what and how was actually measured in the study. There is no serious discussion about the implication, significance, and limits of the results.
In detail, the Title of the paper artificially introduces the concepts of ”sustainable” and ”gymnasium”, which are not supported by the results, nor explored in the literature review, and used in the measurement scheme.
The Abstract does not give clear information about the research design, besides the misleading formulation ”integrated the quantity and quality research approaches” which, besides the incorrect wording, makes the reader assume a mixed-method research design. However, such an approach is not clarified, nor justified in the Material and Methods section. Moreover, the Abstract does not clearly present the main findings, contribution, and implications of the study.
To elicit readers` interest for the paper, the first paragraph from the Introduction needs to be reformulated so that it will become clearer, with less redundant formulations, and too many undeveloped and unrelated ideas (lines 30-31; 36-37). The second paragraph from the Introduction is also ambiguously formulated. In the third paragraph, the author assumes to use Koetler`s spectrum of museum experience as the main theoretical framework (line 48) but it does not make any reference on how it was used or why it was chosen, nor does he structure its literature review or crafting research instruments based on it.
In the Literature review, the author only provides the wide context of the study without digging into justifying the theoretical framework and focusing on showing what is been known about the dimensions used in the study (visceral, cognitive, emotions, etc.) and what further needs to be known (the gap in the literature). Although the sources are relevant and up-to-date, their contribution is not clearly communicated leaving the reader rather confused (lines 58-61; 65-66; 76-78; 82-83). The author criticises scholars for lack of clear conceptual definitions but does the same thing by using various concepts as taken-for-granted (”experience”; "multi-perception"; ”phenomena”; ”embodiment”; ”multi-sensory immersion” etc.) without defining them, or at least producing a working definition from what is already known or showing how the study can contribute to improve the defining process.
The research design is not explicitly presented in the Material and Methods section. The author does not offer enough evidence of acknowledging the difference between qualitative and quantitative methods, or the commonly accepted concepts for research techniques and instruments (for instance by considering observation as a type of survey, by naming the observation grid an observation table, etc.). The choice of research methods, techniques and instruments is not justified, nor their specificity clarified in this section (the reader only finds about various instruments in the Results section, while the questionnaire is only introduced at the end of the paper without being explained, nor anchored in the body of the paper). The measurements are not justified either (is not very clear what concepts were measured, how were they operationalised in dimensions, indicators, and variables based on which research instruments were crafted). The techniques of applying the research instruments, although mentioned in various parts of the paper, are not justified, nor described in so much detail so that the study be reproducible. There is no detail about the specificity of the in-depth interview. The sampling method and procedure are not explained, nor justified. The choice of statistical analysis and formula for an index that is used are not stated, nor justified in this section.
The volume of the sample is only mentioned in the Results and Discussion section (without any rationale behind it). In this section, the author introduces new literature sources (some of them being the framework for building one of the data collection instruments, namely the observation grid). Although the Literature review focuses on the concept of multi-senses, the study also reports results about various types of museum experiences (introspective, social, object) and communication effects (cognitive, emotional) which were not previously defined, nor analysed. The results are mainly reported without seriously discussing them and extracting their significance and implications. There are two frameworks used for two different data collection instruments without making any connection between them, hence not leaving space for the triangulation of results. Although in the Literature review section, the gap is presented to be the lack of proper definitions of concepts related to the sensorial experience of museum visits, the study fails to provide them (or to discuss how the results contribute to bridging this gap). Ignoring the literature review, which is up-to-date but rather unfocused, the author decides (without any offered justification) to frame the paper and craft the data collection tools based on two models from two decades ago. From the content of the paper is impossible for the reader to assess the generalisability of the results as the probability of the sample is not transparent, nor the characteristics of the frame population. The contribution of the study is not clearly stated.
The Conclusions do not entirely support the results. The limitations of the study are missing and also the prospect for future studies.
Although the content of the paper has potential to be of value, the way it was written and structured shows no consideration for the reader, which is not incrementally guided through understanding the research process. All considered I suggest changing the Title, modifying the Abstract, rewriting the Introduction and Literature Review, adding information in the Material and Methods and the Results and Discussion, and rewriting the Conclusions.

Reviewer 3 Report
The article focuses on the role of senses and emotional perception of museums' presentations. The multilayered way od creating museum' exhibition (sensory, aesthetic, emotional, intellectual) experienced in Taizhou Museum was analysed from different points of museological concepts. This innovative method of evaluation of museum activities is the strongest part of this study. Should be wider implemented and disseminated.
What is lacking? Description of results might be wider described, the conclusion is really important for museum practice, for evaluation of museum activity done by it's organizer, and for the public information.
Tables and figures and clear and well constructed.
"Senses in museums" seems to be one of the most important segment of museum' planning. Congratulations to the author to present this subject.

Round 2
Reviewer 2 Report
The authors have satisfactorily responded to my questions and made the necessary changes to the manuscript The revised version of the manuscript is an improved one.